# Prepregnancy Protein Source and BCAA Intake Are Associated with Gestational Diabetes Mellitus in the CARDIA Study

**DOI:** 10.3390/ijerph192114142

**Published:** 2022-10-29

**Authors:** Meghana D. Gadgil, Katherine H. Ingram, Duke Appiah, Jessica Rudd, Kara M. Whitaker, Wendy L. Bennett, James M. Shikany, David R. Jacobs, Cora E. Lewis, Erica P. Gunderson

**Affiliations:** 1Division of General Internal Medicine, Department of Medicine, University of California, San Francisco, CA 94143, USA; 2Department of Exercise Science and Sport Management, Kennesaw State University, Kennesaw, GA 30144, USA; 3Department of Public Health, Texas Tech University Health Sciences Center of Statistics and Analytical Sciences, Lubbock, TX 79409, USA; 4Department of Statistics and Analytical Sciences, Kennesaw State University, Kennesaw, GA 30144, USA; 5Department of Health and Human Physiology, Department of Epidemiology, University of Iowa, Iowa City, IA 52242, USA; 6Division of General Internal Medicine, Department of Medicine, Johns Hopkins School of Medicine, Baltimore, MD 21205, USA; 7Division of Preventive Medicine, School of Medicine, University of Alabama at Birmingham, Birmingham, AL 35294, USA; 8Division of Epidemiology & Community Health, School of Public Health, University of Minnesota, Minneapolis, MN 55455, USA; 9Department of Epidemiology, University of Alabama at Birmingham, Birmingham, AL 35294, USA; 10Division of Research, Kaiser Permanente Northern California, Oakland, CA 94612, USA; 11Department of Health Systems Science, Kaiser Permanente Bernard J. Tyson School of Medicine, Pasadena, CA 91101, USA

**Keywords:** gestational diabetes, diet quality, pregnancy, nutrition, protein

## Abstract

Diet quality and protein source are associated with type 2 diabetes, however relationships with GDM are less clear. This study aimed to determine whether prepregnancy diet quality and protein source are associated with gestational diabetes mellitus (GDM). Participants were 1314 Black and White women without diabetes, who had at least one birth during 25 years of follow-up in the Coronary Artery Risk Development in Young Adults (CARDIA) cohort study. The CARDIA *A Priori* Diet Quality Score (APDQS) was assessed in the overall cohort at enrollment and again at Year 7. Protein source and branched-chain amino acid (BCAA) intake were assessed only at the Year 7 exam (*n* = 565). Logistic regression analysis was used to determine associations between prepregnancy dietary factors and GDM. Women who developed GDM (*n* = 161) were more likely to have prepregnancy obesity and a family history of diabetes (*p* < 0.05). GDM was not associated with prepregnancy diet quality at enrollment (Year 0) (odds ratio [OR]: 1.01; 95% confidence interval [CI] 0.99, 1.02) or Year 7 (odds ratio [OR]: 0.97; 95% confidence interval [CI] 0.94, 1.00) in an adjusted model. Conversely, BCAA intake (OR:1.59, 95% CI 1.03, 2.43) and animal protein intake (OR: 1.06, 95% CI 1.02, 1.10) as a proportion of total protein intake, were associated with increased odds of GDM, while proportion of plant protein was associated with decreased odds of GDM (OR: 0.95, 95% CI 0.91, 0.99). In conclusion, GDM is strongly associated with source of prepregnancy dietary protein intake but not APDQS in the CARDIA study.

## 1. Introduction

Gestational diabetes mellitus (GDM) affects approximately 8% of pregnant women in the United States [1,2] and the prevalence is increasing, especially among women of racial and ethnic minorities [3,4,5,6] and those who begin pregnancy with obesity, older maternal age, and a family history of diabetes. GDM is associated with macrosomia [7], newborn hypoglycemia and Cesarean section delivery [8] and is a major contributor to morbidity for both mother and infant. A pregnancy complicated by GDM is linked to adverse maternal health consequences, including later life hypertension, cardiovascular disease [9,10], and a 7–10 fold higher risk of developing type 2 diabetes in the future [11,12,13]. Due to the rising prevalence of maternal obesity and increasing maternal age, GDM prevalence has increased in all age, race, and ethnic groups at an average annual rate of 3.7% from 2011 through 2019 [14].

Cardiometabolic impairment prior to pregnancy is associated with higher risk for GDM, though this risk may be modified by lifestyle factors. In the prospective Coronary Artery Risk Development in Young Adults (CARDIA) study, prepregnancy cardiometabolic risk factors, including fasting glucose, insulin and LDL cholesterol, were associated with a 2- to 5-fold higher relative risk of GDM independent of prepregnancy body size [15]. Moreover, 26.7% of women with overweight or obesity developed GDM if they had 1 or more additional prepregnancy cardiometabolic risk factors, compared to only 7.4% of those with no additional risk factors [15]. Risk for GDM also was associated with prepregnancy increases in both body weight and waist circumference and lower levels of prepregnancy cardiorespiratory fitness in CARDIA participants [16,17]. Physical activity prior to and during early pregnancy also has been associated with a decreased risk of incident GDM [17]. Thus, modifiable risk factors that are present prior to pregnancy can affect the development of GDM, yet much less is known about specific dietary composition that may contribute to risk.

Diet is a modifiable behavioral risk factor that has been associated with risk of prediabetes, overt diabetes and GDM [18,19,20,21]. A Western dietary pattern, characterized as high in red and processed meats, refined grains and sugar-sweetened beverages, and low in fiber, has been associated with increased risk of GDM in large, longitudinal cohort studies [18,20,22,23]. Similarly, large-scale studies have found higher pregnancy intake of animal-based fat, heme iron and fried foods [20,24] to be associated with GDM. Conversely, dietary patterns characterized by a higher consumption plant-based food products in the prepregnancy period were associated with a lower risk for developing GDM [25,26]. Non-pregnant young adults also were less likely to develop diabetes in middle age if they transitioned to a more plant-centered dietary pattern, as assessed by the plant-centered A Priori Diet Quality Score (APDQS) in CARDIA [27]. However, this diet score has not been studied previously in relation to GDM risk.

Branched chain amino acids (BCAA) have been associated with insulin resistance and metabolic impairments in children and adults, both pregnant and non-pregnant. The three BCAAs—leucine, isoleucine, and valine—are found in a variety of protein sources and have been shown to disrupt glucose homeostasis in skeletal muscle [28]. Animal-based protein sources contain high concentrations of the branched chain amino acids (BCAA), and short-term modification of these food groups has been shown to alter circulating levels of BCAAs in randomized trials [29,30]. Circulating levels of BCAA have been associated with obesity and insulin-resistant states through impaired protein anabolism and impaired insulin signaling [31,32,33]. BCAA added to a high-fat diet caused a disruption in insulin signaling and induced insulin resistance in rats [34]. Moreover, circulating BCAA during early pregnancy is predictive of developing GDM [35,36]. These markers have been found to remain high in the postpartum period after GDM [37] and are further associated with insulin resistance postpartum [38]. These studies suggest that animal-based protein and BCAA intake may be modifiable targets that affect insulin signaling pathways. Additional research is needed to determine whether BCAA in the diet during the prepregnancy period is associated with risk for GDM.

The objectives of this study were to determine if dietary source of protein, BCAA intake, or dietary pattern during the prepregnancy period are associated with risk of GDM in a biracial cohort in the longitudinal CARDIA study.

## 2. Methods

### 2.1. Study Population

CARDIA is a multicenter, longitudinal, observational cohort study in the United States, enrolling Black and White men and women aged 18–30 years from Birmingham, Alabama; Chicago, Illinois; Minneapolis, Minnesota; and Oakland, California. The objective of the CARDIA study was to characterize the development of cardiovascular disease and cardiovascular risk factors in men and women, beginning in young adulthood. Study protocols have been described elsewhere [39]. At baseline (1985–1986), 5115 participants were enrolled, of whom 2787 were women. Self-reported and measured outcomes were collected in follow-up exams at years 2, 5, 7, 10, 15, 20, 25, and 30 years after baseline. Retention rates were 91%, 86%, 81%, 79%, 74%, 72%, 72%, and 71% of the surviving cohort, respectively. The institutional review boards at the University of Alabama at Birmingham, Northwestern University, University of Minnesota, and Kaiser Permanente approved the primary CARDIA study protocols. Written informed consent was obtained at each exam. The CARDIA steering committee approved this secondary analysis.

CARDIA participants who had a live birth between baseline and 25 years of follow-up were included. Live births were defined as delivery of a live infant >20 weeks gestation, conceived after the baseline CARDIA examination (Year 0) and delivered during the 25-year follow-up period between examinations. Of the 2787 women enrolled in CARDIA at baseline, those who had no births during the 25-year follow-up period were excluded (*n* = 1425). Women who were diagnosed with GDM or type 2 diabetes prior to their first birth during the 25-year study period (index pregnancy) were excluded (*n* = 16) [15]. Exclusions were further made for women with missing or implausible caloric intakes (<600 kcal/day or >6000 kcal/day) (*n* = 32).

Participants included 1314 women at baseline (Year 0) who had one or more births during the 25-year study period (Figure 1). From this cohort, 951 births were recorded in the time span from Years 0–7 and 565 births were recorded from Years 7–25. Approximately 31.6% of the participants had one or more births prior to the Year 0 exam. Births prior to the Year 0 exam were not included in the analyses, though all births after Year 0 from eligible parous participants were included. We excluded women with type 2 diabetes or GDM during a pregnancy prior to enrollment at Year 0 (*n* = 16).

Year 7–25 analyses included 565 women from the original cohort who had one or more births during the Year 7–25 interval, with no prior history of GDM or type 2 diabetes. Participants in the Year 7–25 analyses included 514 women without GDM and 51 women with GDM.

### 2.2. Data Collection Instruments

Pregnancy data, including self-reported diagnosis of GDM, were collected at baseline and at each CARDIA follow-up examination. During the data-collection period from enrollment in 1985 (Year 0) through 2011 (Year 25), prevailing clinical diagnostic criteria for GDM initially adhered to the National Diabetes Data Group criteria (1979) [40], and evolved to favor the less restrictive Carpenter and Coustan criteria (1982) [41] based on guidelines from the American Diabetes Association and American College of Obstetrics and Gynecology. Gunderson et al. previously validated the self-report of GDM among 165 women for whom oral glucose tolerance testing results were available [15] using the Carpenter and Coustan criteria [15]. Per this prior investigation, validation was 100% sensitive (20 of 20) and 92% specific (134/145).

The aim of this study is to determine whether prepregnancy diet quality and sources of protein are associated with GDM. Dietary data were collected at the CARDIA Year 0 exam and Year 7 exam and were used to determine associations with development of GDM through exam Year 25. The Year 0 assessment (Diet Quality Cohort) included diet quality score data and was used to analyze associations between diet quality and incidence of GDM in births from Years 0–25. The Year 7 assessment (Protein Source Subset) provided an additional detailed examination of macronutrients and BCAA. This exam was used to determine associations between protein intake and incidence of GDM in births that occurred from Years 7–25 of the CARDIA study.

The CARDIA diet history was interviewer-administered and collected data on the consumption of foods over the previous month, with open-ended questions about foods eaten. The foods were then assigned to 166 food categories using Nutrition Data System for Research (NDSR versions 10, 20, and 36; at CARDIA Year 0, Year 7, and Year 20, respectively; Nutrition Coordinating Center at the University of Minnesota) [42]. The NDSR-generated food categories were compiled into 46 food groups based on nutritional value and culinary usage by CARDIA investigators. Food group intake was determined as the number of servings per day of each food within a food group. The CARDIA A Priori Diet Quality Score (APDQS) was assessed in both Year 0 and Year 7 by classifying food groups as beneficial (20 groups), adverse (13 groups), or neutral (13 groups) based on their hypothesized disease relations, as previously defined in CARDIA [43]. Individual food group intakes were then assigned a consumption score (from 0 to 4), with 0 for non-consumer and 4 for frequent consumption. The APDQS was the sum of food group scores (0 to 4 for beneficial foods and 4 to 0 for adverse foods). Neutral foods were not included in the construction of the APDQS (see Appendix A).

The year 7 analyses (Protein Source) were based on data from the CARDIA interviewer-administered diet history, assessed using an updated, extensive NDSR food and nutrient database, allowing for more specific quantification of protein source and BCAA intake. Animal protein intake is reported as grams per 100 kcal energy intake and defined as all protein from animal sources, including meat, fish, eggs and dairy. Vegetable protein includes all protein from plant-based sources, also reported as grams per 100 kcal energy intake. Total protein intake is determined by summing animal and vegetable protein. The proportion of animal or vegetable protein is reported as percent of total protein intake.

At each examination, social and behavioral characteristics including medication use, alcohol intake, cigarette smoking, education, and physical activity were recorded by standardized structured interviews or questionnaires. Cigarette smoking was categorized as “never”, “past” or “current”. Alcohol consumption was recorded in drinks per week from a self-reported questionnaire. Education was determined by years of schooling. Physical activity was captured by self-report using the CARDIA Physical Activity History [44]. The physical activity score was calculated as the number of months reported for each activity weighted by intensity (estimate of energy expenditure per minute), with increased weight for the number of months with activity greater than the threshold.

### 2.3. Outcome Definition and Statistical Analysis

GDM status was defined for each birth. Individuals were included in the GDM group if at least 1 birth was affected by GDM between consecutive study exams. For continuous variables, *t*-tests were used to compare women with and without GDM for normally distributed variables and Mann–Whitney U tests for variables with skewed distributions. Comparisons of categorical variables between women with and without GDM were conducted using chi-square tests. APDQS was converted to tertiles for analysis.

Logistic regression was used to evaluate the associations between dietary factors and odds of GDM. Models were adjusted for: age, parity, physical activity, energy intake, cholesterol and saturated fat intake and BMI, included as continuous variables, and race and family history of diabetes included as categorical variables, selected a priori as confounders based on prior data and biological plausibility. Models were also adjusted for time from CARDIA exam to delivery date. No interaction effect was found between APDQS and BMI (*p* = 0.77). SAS version 9.4 (SAS Institute, Inc., Cary, NC, USA) was used for all analyses.

## 3. Results

Baseline characteristics of women who subsequently developed GDM (*n* = 161) and women who did not develop GDM (*n* = 1153) are displayed in Table 1 (Diet Quality Cohort) and Table 2 (Protein Source Subset). Approximately half of all women self-identified as Black in both GDM and non-GDM groups. No baseline differences were found in age, physical activity, education, smoking status, or parity between women with or without subsequent GDM. However, women with a GDM birth were more likely to be classified as overweight or obese by BMI, as compared to women without GDM. They also were more likely to have a family history of diabetes and higher fasting glucose and insulin levels, though these were within the normal range. In the Protein Source Subset at baseline (Year 7), most findings were similar to those observed in the larger Diet Quality group assessed at Year 0. However, a higher total cholesterol level, lower HDL cholesterol level, higher fasting glucose levels within the normal range, and no difference in BMI or insulin levels were observed between women who developed GDM (*n* = 51) and women who did not (*n* = 514) (Table 2). No significant difference was found in prepregnancy APDQS between women with GDM and without GDM in either the Diet Quality Cohort (Table 1) or in the Protein Source Subset (Table 2). In the latter, intake of protein from animal sources and dietary cholesterol were greater among women with GDM, while proportion of protein intake from vegetable sources was lower. There was no difference observed between groups in mean BCAA intake.

Logistic regression models controlling for age, family history, race, parity, physical activity, total daily energy intake, time to delivery and BMI were constructed to evaluate the association of the primary exposure of APDQS with odds of GDM. The APDQS prior to pregnancy was not significantly associated with GDM in the Diet Quality Cohort (Table 3) or in the Protein Source Subset (Table 4).

Additional analyses of protein source were completed in the Protein Source Subset using models controlling for the same covariates listed above, plus overall diet quality and intake of cholesterol and saturated fats. Animal protein intake, as a proportion of total protein intake, was directly associated with odds of GDM (OR: 1.06; 95% CI 1.02, 1.10), Vegetable protein intake, as a proportion of total protein intake, also was inversely associated with odds of GDM (OR: 0.95; 95%CI 0.91, 0.99). Conversely, GDM was not significantly associated with either animal protein or vegetable protein intake when these were expressed as grams per 100 kcal total intake. Total BCAA intake was not significantly associated with odds of GDM, while BCAA intake as a proportion of protein intake was associated with odds of GDM (odds ratio [OR]: 1.57; 95% CI 1.02, 2.40). These findings are displayed in Table 4.

## 4. Discussion

Prepregnancy diet is a significant determinant of subsequent GDM risk in the CARDIA Study. Our findings highlight the importance of prepregnancy dietary protein source in determining subsequent risk for GDM, independent of energy intake, diet quality score, BMI, physical activity, family history of diabetes, dietary cholesterol and saturated fat. A 1% higher intake of prepregnancy animal protein, as a proportion of total protein, was associated with a 6% increased odds of developing GDM. Higher prepregnancy intake of BCAAs, as a proportion of total protein intake, was similarly associated with increased odds of subsequent GDM. Conversely, a 1% higher proportion of plant-based protein to total protein intake was associated with 5% lower odds of developing GDM in the CARDIA Study.

Dietary protein and its sources have been the focus of other studies as potential modifiable risk factors for GDM [18]. Similar associations have been reported between a high-meat dietary pattern (i.e., Western pattern) and increased risk for GDM [45], but this study is the first to show that proportion of protein consumption from animal versus plant sources and BCAA in the prepregnancy period are associated with subsequent GDM in a biracial cohort. These associations are independent of dietary quality score, saturated fat intake, and dietary cholesterol.

### 4.1. Animal Proteins and GDM

Our investigation revealed that a higher intake of animal protein relative to total protein prior to pregnancy was associated with higher odds of GDM in the CARDIA Study, while a higher proportion of vegetable protein was associated with lower odds of GDM. Prior longitudinal cohort studies have reported similar relationships. In the Nurses’ Health study, a cohort of predominantly White women, greater prepregnancy intake of protein and fat from animal sources were each associated with a higher incidence of GDM, independent of BMI and other factors [18,20]. A study of an Australian population found an association between a “Meats, snacks, and sweets” diet pattern and GDM risk, though the association was attenuated when controlling for BMI [22]. In the current analysis of the CARDIA cohort, the association between GDM risk and higher proportion of animal protein intake persisted with adjustments for saturated fat, cholesterol, and known clinical and lifestyle risk factors for GDM, including BMI, maternal age, family history of diabetes and physical activity.

### 4.2. Plant-Based Protein

Increased prepregnancy consumption of plant-based protein, as a proportion of total protein intake, was associated with lower odds of GDM in CARDIA. These findings are consistent with previous longitudinal studies reporting a decreased risk of GDM with higher plant-based protein consumption [18,45,46]. In the Nurses’ Health Study, a model that replaced 5% animal protein with vegetable protein resulted in a decreased risk of GDM by 51% [18]. In our study, prepregnancy intake of vegetable protein, expressed per 100 kcal total energy intake, was not significantly associated with GDM, suggesting that the protein source, but not the protein quantity, is most closely related to glycemic dysfunction in pregnancy. Plant-based foods contain a wide range of other non-protein compounds which may contribute to the negative association with risk for GDM, such as dietary fiber [47]. However, dietary fiber intake at baseline (Year 0) was not different in the participants who developed GDM than in those who did not. In the Protein Source Subgroup analyses, dietary fiber was slightly lower in those who later developed GDM, but this difference was not significant. To minimize the influence of fiber and other known, and unknown, dietary factors from plant-based diets on GDM risk, our analyses were adjusted for the CARDIA A Priori Diet Quality Score (APDQS), which emphasizes a wide range of plant-based foods.

### 4.3. BCAA and GDM

In CARDIA, the prepregnancy dietary intake of BCAA, expressed as a proportion of total protein intake, was associated with GDM, though total BCAA intake as grams per 100 total kcal was not. This suggests that sources of protein high in BCAA, specifically animal protein-based foods, may have a particular metabolic influence on risk of GDM, though determination of causation and associated mechanisms will require intervention studies. Dietary sources of BCAA include a wide range of foods, from red and processed meats to certain fish, as well as nuts such as walnuts. These foods all have variable associations with glucose intolerance and diabetes in longitudinal studies, and warrant further research focused on food source of BCAA and association with metabolic disease [31,48,49,50,51]. Both dietary intake and circulating levels of BCAA have been associated with obesity, insulin resistance, diabetes and/or GDM in previous studies [29,30].

Previous studies suggest that BCAA leads to insulin resistance by interfering with insulin signaling pathways. Circulatory levels of the BCAA were strongly associated with insulin resistance, as indicated by glucose disposal rate after hyperinsulinemic-euglycemic clamp. This effect was particularly apparent in women [33]. In a rat feeding model, BCAA dietary overload led to disruption of insulin signaling, but only when BCAA was combined with a high-fat diet [34], as can be commonly found in a typical Western-style diet high in animal-based products. The accumulation of incompletely oxidized lipid-derived metabolites pointed to an overload of mitochondrial fuel oxidation in muscle, while further examination of the insulin signaling pathway suggests that chronic mTOR/S6K1 kinase pathway activation is the likely culprit [34].

### 4.4. Dietary Patterns

Prior observational studies have reported inverse associations between incident GDM and various indicators of prepregnancy dietary quality, such as the prudent dietary pattern, the alternate Healthy Eating Index, the alternate Mediterranean diet or the Dietary Approaches to Stop Hypertension and a plant-based diet index, all of which emphasize a focus on plant-based foods [45,52,53]. Similarly, the CARDIA APDQS emphasizes a plant-based diet and is inversely associated with risk of type 2 diabetes in the larger CARDIA population [27,54]. However, the relationship between prepregnancy APDQS and subsequent risk of GDM was not significant, when controlling for maternal age, race, parity, family history of diabetes, overall energy intake, physical activity, time to delivery, and prepregnancy BMI. This finding contradicts our primary hypothesis and previous findings from other cohort studies [52], but most likely is due to calculation differences in diet quality scores.

It is possible that the inclusion of a more diverse patient population alters the association between diet quality and GDM. While race was not a statistically significant covariate, differences were observed between Black and White participants at baseline in vegetable protein consumption, energy intake and APDQS, though analysis of effect modification was not performed on the small sample of GDM cases.

### 4.5. Strengths and Limitations

One of the major strengths of this investigation is the longitudinal collection of diet data in a Black and White cohort of young adults that allow a priori assessment with future GDM. This study also has some limitations. First, only 51 GDM pregnancies were reported during Years 7–25 and included in the Protein Source Subset, out of 565 women who had births during this timeframe. The low number of incidents during this timeframe limited the power to detect our main effects of protein source and BCAA on risk of GDM. Nevertheless, the associations between protein source and GDM risk were significant, indicating a robust effect among these findings. We found associations between GDM and the proportional intake of animal vs. vegetable-based protein, rather than the total amount of these nutrients. Thus it is not possible to identify an optimal quantity of vegetable-based protein to prevent GDM through this cohort study. Rather, our findings suggest that it is beneficial to consume proportionally more plant-based protein than animal-based protein in the prepregnancy period.

To further test the lack of association between APDQS and GDM Risk, we analyzed this relationship both on the larger Diet Quality Cohort and on the smaller Protein Source Subgroup. Second, the dietary data was collected in CARDIA several years before pregnancy. It is possible that dietary intake and quality may have changed over time or during the immediate prepregnancy period, or that other factors, such as differences in prepregnancy cardiometabolic risk status (high fasting insulin, impaired fasting glucose, and low HDL-C), BMI or central obesity, influenced the overall risk of GDM, as previously reported [15]. Indeed, national trends show a small increase in diet quality in adults over time [55]. Consistent with this, we observed a slight increase in diet quality among both groups between the baseline Year 0 and the Year 7 exam. There was no statistically significant interaction between APDQS and BMI in this analysis. While baseline levels of cardiometabolic risk factors were different between women who developed GDM and those who did not, average values were within normal limits in both groups. Even so, our analyses were adjusted for time to delivery to minimize any influence of time variability. Thus, the data collected in this cohort study are robust and the strength of the study remains the long-term follow-up of participants.

Our findings strengthen current evidence that a prepregnancy diet including a higher proportion of plant-based protein and a lower proportion of animal-based protein is associated with a reduced odds of developing GDM in a cohort of Black and White women. We further show an increased risk for GDM with higher dietary intake of BCAA during prepregnancy. Therefore, prepregnancy nutrition counseling that includes shifting protein sources from animal sources toward plant sources may be a simple and effective tool to reduce risk for GDM. Further investigation is needed to study the impact of replacing animal protein with nutrient-dense plant foods on the development of GDM.

## Figures and Tables

**Figure 1 ijerph-19-14142-f001:**
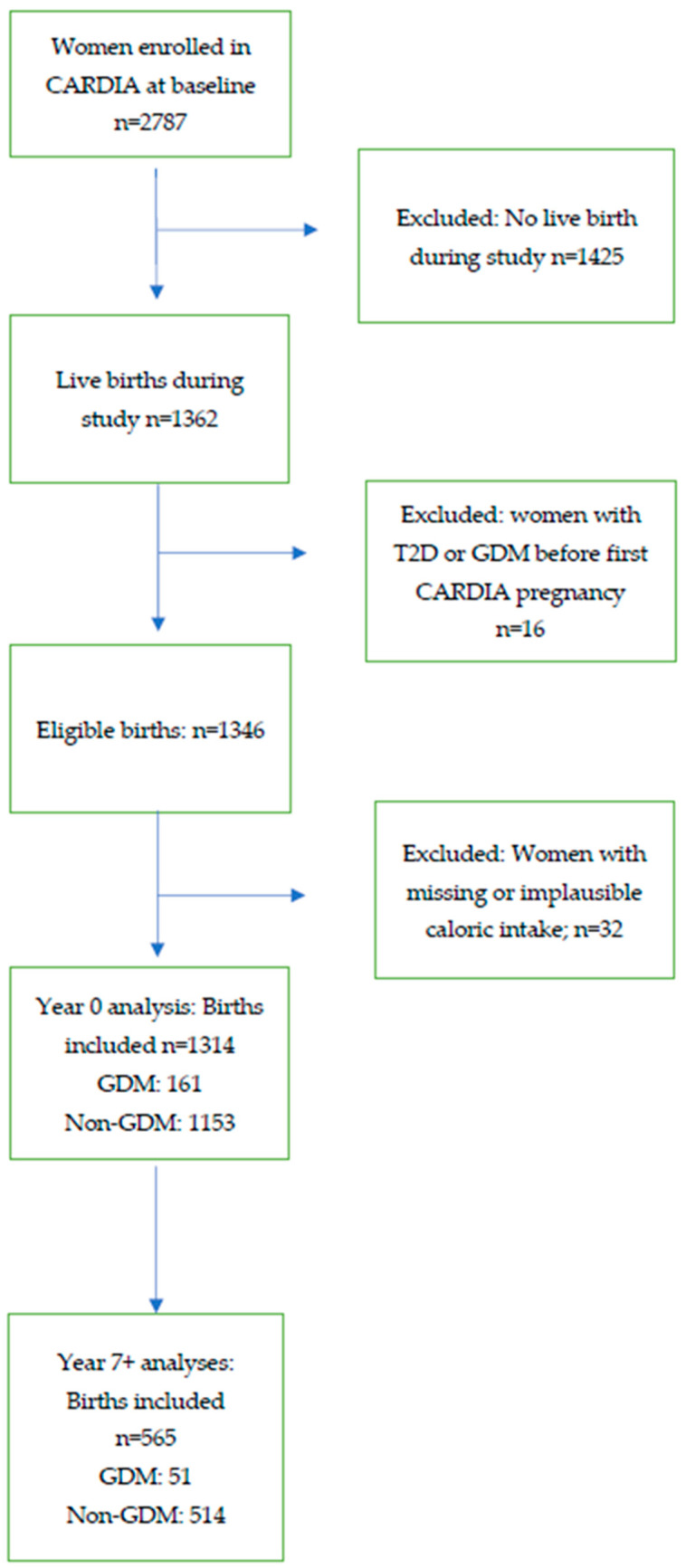
Flow diagram of analytic sample selection, the Coronary Artery Risk Development in Young Adults (CARDIA) Study, 1985–2011.

**Table 1 ijerph-19-14142-t001:** Diet Quality Cohort. Participant characteristics at baseline (Year 0) by subsequent GDM pregnancy status, the CARDIA study (*n* = 1314), 1985–2011.

Characteristics	GDM Pregnancies (*n* = 161)	Non-GDM Pregnancies (*n* = 1153)	*p* Value ^a^
Age, years, mean (SD)	24.5 (3.9)	24.1 (3.6)	0.236
Race, Black, *n* (%)	78 (48.4)	576 (50.0)	0.737
>High school education, *n* (%)	110 (68.3)	749 (65.0)	0.427
Current smoker, *n* (%)	49 (30.4)	277 (24.0)	0.080
Family history of diabetes, *n* (%)	33 (20.5)	146 (12.7)	0.010
Body mass index, *n* (%)			0.003
Normal weight	97 (61.8)	810 (72.8)	
Overweight	30 (19.1)	187(16.8)	
Obese	30 (19.1)	116 (10.4)	
Nulliparous, *n* (%)	107 (66.5)	792 (68.7)	0.586
Physical Activity Score	292 (166–460)	295 (154–476)	0.970
Diet Quality Score	65.0 (12.4)	63.8 (13.7)	0.315
Energy Intake, kcal ^b^	1993 (1553–2696)	2089 (1623–2821)	0.118
Protein, g/100 kcal	3.8 (0.7)	3.7 (0.7)	0.140
Carbohydrate, g/100 kcal	11.6 (1.8)	11.7 (1.8)	0.448
Fat, g/100 kcal	4.2 (0.7)	4.2 (0.7)	0.574
Polyunsaturated Fat, g/100 kcal ^b^	0.75 (0.65–0.89)	0.74 (0.62–0.90)	0.593
Saturated Fat, g/100 kcal	1.6 (0.4)	1.6 (0.3)	0.606
Fiber, g/100 kcal ^b^	0.20 (0.15–0.27)	0.20 (0.15–0.27)	0.660
Cholesterol, mg/100 kcal ^b^	16.2 (12.8–20.2)	15.4 (12.0–19.4)	0.051
Alcohol, g/100 kcal ^b^	0.15 (0.00–0.44)	0.13 (0.01–0.41)	0.861
Fasting glucose, mg/dL	81.2 (9.4)	79.1 (7.4)	0.009
Insulin, uU/ML	11.0 (9.0–15.0)	10.0 (8.0–13.0)	0.004
Triglycerides, mg/dL	64.0 (51.0–81.0)	57.0 (43.0–77.0)	<0.001
HDL-Cholesterol, mg/dL	54.9 (13.6)	56.3 (12.7)	0.220

*p*-value testing for differences between groups using chi-square tests, independent samples *t*-tests, Data presented at mean (standard deviation) unless otherwise indicated. ^a^ *p*-value testing for differences between groups using independent samples *t*-tests or Mann–Whitney U tests, as appropriate. ^b^ Median (interquartile range).

**Table 2 ijerph-19-14142-t002:** Protein Source Subset. Participant characteristics at Year 7 by subsequent GDM pregnancy status, the CARDIA study (*n* = 565), 1992–2011.

Characteristics	GDM Pregnancies (*n* = 51)	Non-GDM Pregnancies (*n* = 514)	*p* Value ^a^
Age, years, ^b^	30.0 (26.0–33.0)	30.0 (27.0–33.0)	0.369
Race, Black, *n* (%)	21 (41.2)	213 (41.4)	0.971
>High school education, *n* (%)	39 (76.5)	421 (81.9)	0.347
Current smoker, *n* (%)	13 (25.5)	103 (20.0)	0.365
Family history of diabetes, *n* (%)	13 (25.5)	72 (14.2)	0.041
Body mass index, *n* (%)			0.122
Normal weight	25 (52.1)	292 (62.5)	
Overweight	8 (16.7)	87 (18.6)	
Obese	15 (31.1)	88 (18.8)	
Nulliparous, *n* (%)	27 (52.9)	284 (55.3)	0.770
Physical Activity Score	204 (84- 358)	219 (115–406)	0.915
Diet Quality Score	66.5 (13.4)	69.8 (11.5)	0.053
Energy Intake, kcal ^b^	2201 (1725–2680)	2134 (1675–2901)	0.777
BCAA, g/100 kcal ^b^	15.2 (11.7–17.7)	14.5 (10.8–18.8)	0.430
Protein, g/100 kcal	3.8 (0.7)	3.7 (0.7)	0.336
Vegetable Protein, g/100 kcal ^b^	1.2 (1.0–1.3)	1.3 (1.1–1.5)	0.013
Veg. Protein/Total Protein, % ^b^	30.1 (26.5–35.8)	34.1 (27.8–40.2)	0.006
Animal Protein, g/100 kcal	2.7 (0.7)	2.4 (0.7)	0.033
Animal Protein/Total Protein, % ^b^	69.2 (63.6–72.6)	64.6 (58.8–71.1)	0.003
Glycine/Total protein, %	4.0 (0.4)	3.9 (0.5)	0.856
Carbohydrate, g/100 kcal	12.3 (1.4)	12.8 (1.9)	0.032
Fat, g/100 kcal ^b^	4.1 (3.7–4.3)	3.9 (3.3–4.3)	0.024
Polyunsaturated Fat, g/100 kcal ^b^	0.78 (0.62- 0.91)	0.74 (0.61–0.90)	0.490
Saturated Fat, g/100 kcal	1.4 (0.3)	1.3 (0.3)	0.014
Fiber, g/100 kcal ^b^	0.76 (0.62–0.90)	0.87 (0.70–1.08)	0.010
Cholesterol, mg/100 kcal ^b^	13.0 (10.5–15.2)	11.1 (8.5–13.9)	0.003
Alcohol, g/100 kcal ^b^	0.05 (0.01–0.18)	0.10 (0.00–0.39)	0.158
Blood test parameters:			
Insulin, uU/ML ^b^	11.0 (9.0–15.0)	11.0 (8.0–14.0)	0.210
Fasting glucose, mg/dL	87.5 (9.1)	84.5 (7.8)	0.014
Triglycerides, mg/dL ^b^	66.0 (47.0–91.0)	59.0 (43.0–86.0)	0.277
HDL-Cholesterol, mg/dL	51.9 (11.8)	58.0 (13.7)	0.003

^a^ *p*-value testing for differences between groups using independent samples *t*-tests or Mann–Whitney U tests. ^b^ Median (interquartile range).

**Table 3 ijerph-19-14142-t003:** Diet Quality Cohort. Odds ratios (OR) and 95% confidence intervals (CI) for the association of CARDIA a priori Diet Quality score (APDQS) with incident GDM (*n* = 1314), 1985–2011 *^,†^.

Dietary Characteristics	OR (95% CI)	*p* Value
Diet quality		0.107
First tertile	1	
Second tertile	1.66 (1.04–2.65)	
Third tertile	1.45 (0.83–2.54)	
Per 1-point increase	1.01 (0.99–1.02)	0.461
Per 1 standard deviation (13.5)	1.09 (0.87–1.37)	0.461

* Adjusted for maternal age, race, parity, family history of diabetes, energy intake, physical activity, time to delivery, and prepregnancy BMI. ^†^ CARDIA a priori Diet Quality score (APDQS) rated 0–132, with 0 as lowest quality and 132 as highest quality diet.

**Table 4 ijerph-19-14142-t004:** Odds ratios (OR) and 95% confidence intervals (CI) for the association of dietary characteristics with incident GDM in the Year 7 subset (*n* = 565), 1992–2011 *.

Dietary Characteristics	OR (95% CI)	*p* Value
BCAA	1.08 (0.96–1.20)	0.195
BCAA/total protein	1.57 (1.02–2.40)	0.040
Animal protein	1.02 (1.00–1.04)	0.063
Animal protein/total protein	1.06 (1.02–1.10)	0.006
Vegetable protein	0.95 (0.91–1.00)	0.063
Vegetable protein/total protein	0.95 (0.91–0.99)	0.014
Glycine/total protein	1.03 (0.50–2.10)	0.944
Diet quality ^†^		0.081
First tertile	1	
Second tertile	0.41 (0.18–0.93)	
Third tertile	0.53 (0.23–1.20)	
Per 1-point increase	0.97 (0.94–1.00)	0.060
Per 1 standard deviation (11.8)	0.70 (0.48–1.02)	0.060

* Adjusted for maternal age, race, parity, family history of diabetes (assessed at year 5), energy intake, physical activity, prepregnancy BMI, time to delivery, diet quality and intake of cholesterol and saturated fats. ^†^ Adjusted for maternal age, race, parity, family history of diabetes (assessed at Year 5), energy intake, physical activity, prepregnancy BMI.

## Data Availability

The data that support the findings of this study are available from CARDIA study (https://www.cardia.dopm.uab.edu/) but restrictions apply to the availability of these data, which were used under license for the current study, and so are not publicly available. Data are however available from the authors upon reasonable request and with permission of the CARDIA Steering Committee.

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
