# Peer review of "Prepregnancy Protein Source and BCAA Intake Are Associated with Gestational Diabetes Mellitus in the CARDIA Study"

_ijerph, 2022, doi:10.3390/ijerph192114142_

Round 1

Reviewer 1 Report

In this study, the author evaluated if the dietary source of protein and BCAA intake during the prepregnancy period impacted gestational diabetes. The introduction is good and indicates the purpose of this study. The method is appropriate; figure 1 helps to understand the inclusion of patients in this study.

The tables are helpful in understanding the results with a glance.

This study's findings are subjective, making it challenging to implement in clinical settings. The author should consider adding an amount of vegetarian protein required to prevent gestational diabetes rather than saying a higher amount of vegetarian protein and a lower portion of animal protein are associated with reduced odds of gestational diabetes. Similarly, the author should try to quantify the proportion of BCAA that will increase the risk of gestational diabetes.

Reviewer 2 Report

Manuscript ID: ijerph-1960651

Title: Prepregnancy Protein Source, not Diet Quality, is Associated 2 with Gestational Diabetes Mellitus in the CARDIA Study

Authors: Meghana D. Gadgil, Katherine H. Ingram, Jessica Rudd, Duke Appiah, Kara M. Whitaker, Wendy L. Bennett, James M. Shikany, David R. Jacobs, Jr, Cora E. Lewis, Erica P. Gunderson

 In this retrospective analysis of the associations between diet quality and protein source in the development of GDM among women participants of the longitudinal CARDIA study, the authors attempted to determine if the dietary source of protein, BCAA intake, or dietary pattern during the pre-pregnancy period are associated with risk of GDM in a biracial cohort in the CARDIA study. Using pre-pregnancy data from self-reported diagnosis of GDM that were collected at baseline and at each CARDIA follow-up examination, the authors concluded that a pre-pregnancy diet high in plant-based protein and low in animal-based protein is associated with a reduced odds of developing GDM in a cohort of Black and White women participants of the CARDIA study. Interestingly, the authors provided statistical evidence of a significantly increased risk of GDM with higher dietary intake of BCAA during pre-pregnancy and suggested pre-pregnancy nutritional counselling that includes shifting protein sources from animal sources toward plant sources may be a simple and effective tool to reduce the risk for GDM. The study was well conducted, and the statistical analyses were appropriate and informative. However, further edits and justifications are required conducive to have a meaningful article of this calibre.

Major concerns:

Introduction:

Lines 94-97 and lines 113- 115: redundancy; please consider removal or rephrasing.

Lines 108- 110 and lines 79- 83: redundancy; please consider rephrasing accordingly.

Methods:

Lines 136- 137: redundant with lines 145- 146; please consider removal or rephrasing.

Line 152: Please check if the number of participants (n) in Year 7-25 is 956? Figure 1 dictates n= 565! Please ensure consistency.

Lines 140- 146: Please provide justifications for selecting the Year 0, and Year 7-25 time points in cohorts’ selection and data collection and analyses.

Lines 170- 173: it’s unclear why the authors used dietary data from Year 7 assessment, and not Year 0 for the assessment of the Protein Source Subset in their analysis? what about the associations between protein intake and incidence of GDM in births that occurred from Year 0- 7 of the CARDIA study? Please elaborate with explanations and provide justifications.

Results:

Lines 232- 234: the authors reported a lack of significance in fasting glucose levels between women who developed GDM (n=51) and women who did not (n= 956). However, there are no data on fasting glucose in Table 2? Please provide the missing information or consider rephrasing accordingly.

Reviewer 3 Report

Prepregnancy Protein Source, not Diet Quality, is Associated with Gestational Diabetes Mellitus in the CARDIA Study
Main observations:
The manuscript topic is consistent with the journal content.
 The discussion is consistent with the evidence and arguments and addresses the primary objective.
The authors rather right concluded (based on statistics research) that animal protein intake, as a proportion of total protein intake, was directly associated with odds of Gestational Diabetes Mellitus. Still, conversely, Gestational Diabetes Mellitus was not significantly associated with either animal protein or vegetable protein intake when these were expressed as grams per 100 kcal total intake.
The main conclusion from the manuscript stands out: A 1% higher intake of prepregnancy animal protein, as a proportion of total protein, was associated with a 6% increased odds of developing Gestational Diabetes Mellitus and a 1% higher proportion of plant-based protein to total protein intake was associated with 5% lower odds of developing Gestational Diabetes Mellitus.
Literature is relatively out of date - more than 70% are articles more than five years old - should use more everyday items.
Lack of precision in the name of used statistical test (Wilcoxon-Mann Whitney tests ( METHODS) and Mann Wilcoxon-Mann Whitney tests (under TABLE 1), Mann Whitney U tests (under TABLE 2).
I believe this study would be a candidate for publication in Int. J. Environ. Res. Public Health as an original article, with minor revisions.

Round 2

Reviewer 2 Report

Thank you for your revisions.